# Effects of load on good morning kinematics and EMG activity

Andrew David Vigotsky[1], Erin Nicole Harper[1], David Russell Ryan[1] and Bret Contreras[2]

[1] Kinesiology Program, School of Nutrition and Health Promotion, College of Health Solutions, Arizona State University, Phoenix, AZ, USA
[2] School of Sport and Recreation, Auckland University of Technology, Auckland, New Zealand

## ABSTRACT

Many strength and conditioning coaches utilize the good morning (GM) to strengthen the hamstrings and spinal erectors. However, little research exists on its electromyography (EMG) activity and kinematics, and how these variables change as a function of load. The purpose of this investigation was to examine how estimated hamstring length, integrated EMG (IEMG) activity of the hamstrings and spinal erectors, and kinematics of the lumbar spine, hip, knee, and ankle are affected by changes in load. Fifteen trained male participants (age = 24.6 ± 5.3 years; body mass = 84.7 ± 11.3 kg; height = 180.9 ± 6.8 cm) were recruited for this study. Participants performed five sets of the GM, utilizing 50, 60, 70, 80, and 90% of one-repetition maximum (1RM) in a randomized fashion. IEMG activity of hamstrings and spinal erectors tended to increase with load. Knee flexion increased with load on all trials. Estimated hamstring length decreased with load. However, lumbar flexion, hip flexion, and plantar flexion experienced no remarkable changes between trials. These data provide insight as to how changing the load of the GM affects EMG activity, kinematic variables, and estimated hamstring length. Implications for hamstring injury prevention are discussed. More research is needed for further insight as to how load affects EMG activity and kinematics of other exercises.

## INTRODUCTION

Training specificity, defined as "how exercise programs must match functional activities to produce the greatest performance gains" (*Morrissey, Harman & Johnson, 1995*), is a well-known and researched variable of exercise programming (*Behm & Sale, 1993*; *Campos et al., 2002*; *Graves et al., 1990*; *Graves et al., 1989*; *Izquierdo et al., 2002*; *Morrissey, Harman & Johnson, 1995*). This means that exercises that better mimic the kinetics and kinematics of the movement attempting to be improved, for example, sprinting, will have a greater and more beneficial carryover. Factors often considered when programming for specificity include range of motion, velocity, and type of contraction, but load and the result of load on a movement are less investigated.

Knowing how load affects movement kinematics and electromyography (EMG) activity allows the role of load in resistance training to not only control for intensity, but to also

Corresponding author
Andrew David Vigotsky,
avigotsk@asu.edu

improve specificity, as certain loads may better mimic other movements, especially those in sport. *Kellis, Arambatzi & Papadopoulos (2005)* found that as Smith-machine barbell load increased for the concentric squat from 7 to 70% of one repetition maximum (1RM), hip, knee, and ankle angles were significantly affected. In *Hay et al. (1983)*, it was revealed that there is a progressive increase in trunk inclination; that is, the trunk being more horizontal, at the start of the barbell squat as external load is increased. This change in trunk inclination resulted in the hip extensors assuming markedly more load and the knee extensors markedly less, as determined by resultant joint torques, and so much so that 40% of participants' four repetition maximum yielded greater and equal resultant knee torques than 80 and 60%, respectively (*Hay et al., 1983*). Similar effects were seen in *Bryanton et al. (2012)*, which measured the effects of squat depth and load on relative muscular effort (RME), or, "the muscle force required to perform a task relative to the maximum force the muscle can produce." Knee extensor RME increased with depth, but not load; ankle plantar flexor RME increased with load, but not depth; and hip extensor RME increased with both depth and load. In an isometric bench press, *Pinto et al. (2013)* found that EMG activity of the pectoralis major increased from 60 to 80% of maximum voluntary isometric force production, but the increase from 80 to 90% was not statistically significant. *Aspe & Swinton (2014)* compared the effects of relative loading (60, 75, and 90% of three repetition maximum (3RM)) in the back and overhead squat on kinetics and EMG activity. Although investigators did not run statistical analyses on the effects of load, not all mean EMG values of the muscles measured increased linearly with load. It is clear that the relationship between load, kinematics, kinetics, and EMG is not fully understood.

The good morning (GM) is a popular exercise in the strength and conditioning community for increasing hamstring and spinal erector strength (*Kraemer, Clark & Schmotzer, 1982*), but little research exists on its kinematics and EMG activity. The GM is performed by placing a barbell on one's upper trapezius, slightly higher than the barbell positioning in a back squat, with feet about shoulder width apart, and bending forward from the hips until the trunk is approximately parallel with the floor, followed by hip extension to return to starting position (*Kraemer, Clark & Schmotzer, 1982*). Eccentric hamstring loading has been shown to be beneficial for decreasing the risk of hamstring injury in sport (*Askling, Karlsson & Thorstensson, 2003*; *Croisier et al., 2008*; *Heiderscheit et al., 2010*; *Petersen & Hölmich, 2005*) and increasing the optimum length of the hamstrings (*Brockett, Morgan & Proske, 2001*). Furthermore, it is recommended that trainees progressively increase the length at which muscles are trained in order to reduce the risk of injury (*Malliaropoulos et al., 2012*). Therefore, the GM may be an efficacious exercise in preventing hamstring injury.

The biomechanics of a GM are similar to that of the stiff leg deadlift (SLDL). Both exercises are used to target the hamstrings muscle group (semitendinosus, semimembranosus, and biceps femoris) via an eccentric muscle action during hip flexion and a concentric muscle action during hip extension (*Ebben, 2009*; *McAllister et al., 2014*; *McGill et al., 2009*; *Wright, DeLong & Gehlsen, 1999*). During an SLDL, the load is held in the hands and lifted off the ground, while a GM has the load resting across the shoulders and is

stabilized by the hands. EMG activity of the hamstrings in the GM is similar to that of the SLDL, and the GM may be a good alternative closed kinetic chain movement with a lower quadriceps-to-hamstring ratio to emphasize the hamstrings, or a suitable substitution if gripping the bar is an issue (*Ebben, 2009*).

In addition to training for hamstring strengthening, the GM is utilized to target back extensor muscles (*Kraemer, Clark & Schmotzer, 1982*). *Burnett, Netto & Beard (2002)* noted that peak spinal erector forces in the GM (5929.6 N), while utilizing a typical training load, are comparable to that in the clean, snatch, Romanian deadlift, and bent-over row. *Schellenberg et al. (2013)* investigated the kinetics and kinematics of the good morning and found maximal segmental flexion angles of 5.3, 75.3, and 23.8° for the knee, hip, and lumbar spine, respectively, and a mean normalized L4/L5 moment of 2.75 Nm/BW, but the load used was only 25% of bodyweight. *McGill et al. (2009)* examined the kinematics and EMG activity during unloaded GM and found that participants achieved, on average, 55° of hip flexion and 20° of lumbar flexion, and the greatest muscular activity was seen in the thoracic erector spinae (~17% of maximum voluntary isometric contraction (MVIC)), followed by the lumbar erector spinae (~12.5% MVIC). Biceps femoris activity measured ~5% MVIC. *McAllister et al. (2014)* reported that erector spinae activity in the GM is similar to that in the Romanian deadlift and prone leg curl, but significantly less than the EMG activity recorded in the glute-ham raise. However, it is difficult to digest and analyze data presented by *McAllister et al. (2014)*, as EMG was not normalized.

Despite the GM being a popular exercise in the strength training community, there is a paucity of data on its kinematics and EMG activity (*Ebben, 2009*; *McAllister et al., 2014*; *McGill et al., 2009*; *Schellenberg et al., 2013*). The effects of load during exercise on kinematics and EMG activity have not been rigorously investigated together. The purpose of this study is to investigate how load affects range of plantar flexion, knee flexion, hip flexion, and lumbar flexion motion, in addition to EMG activity of the thoracic erector spinae, lumbar erector spinae, medial hamstrings, and lateral hamstrings during the GM. Lastly, the approximate length of each hamstring muscle at the bottom of the movement will be calculated using the coefficients provided by *Hawkins & Hull (1989)*, wherein investigators modeled several lower extremity muscles during various hip, knee, and ankle flexion positions, and created regression equations that can be utilized to obtain muscle lengths based on sagittal plane joint angles.

## MATERIALS & METHODS

### Experimental approach to the problem

The purpose of this study was to examine the effects of load on movement kinematics and EMG activity during the GM. During the course of one session, participants warmed up, performed a submaximal 1RM estimation test, and performed one repetition of the GM with varying loads (50, 60, 70, 80, and 90% 1RM) in a randomized order, with two minutes rest between each repetition. We hypothesized that as load increases, medial and lateral hamstrings integrated EMG (IEMG) activity would increase, then plateau from 80 to 90%,

similar to the findings of *Pinto et al. (2013)*, and lumbar and thoracic erector spinae IEMG activity would increase due to the increase in L5/S1 torque. In order to compensate and decrease hip extension torque, it was also hypothesized that angle of peak ankle plantar flexion would decrease, angle of peak knee flexion would increase, angle of peak lumbar flexion would increase, and angle of peak hip flexion would decrease, and as a result, mean hamstring length would decrease.

## Subjects

Fifteen healthy men ($n = 15$; age $= 24.6 \pm 5.3$ years; body mass $= 84.7 \pm 11.3$ kg; height $= 180.9 \pm 6.8$ cm) with $8.6 \pm 5.5$ years of weightlifting experience, recruited by word of mouth, participated in this study. All participants had weight-trained consistently for at least two years prior to this investigation. All participants were experienced with the GM, having had performed it on a minimum of 12 different occasions over the 12 months prior to testing. All participants were healthy and denied the existence of any current musculoskeletal or neuromuscular injuries, pain, or illnesses; if one had been discovered during testing, that participant was excluded. All participants filled out an Informed Consent and Physical Activity Readiness Questionnaire (PAR-Q) before beginning. Any participant that answered "Yes" to any of the questions on the PAR-Q was excluded. Participants were advised to refrain from resistance training that targeted the lower body or back for 72 h prior to testing. Using only the barbell on the first warm-up set, participants' form was evaluated to ensure the movement was comfortable and acceptable. If a participant reported pain, discomfort, or failed to perform the movement correctly, that participant was excluded. If, for any reason, a participant could not complete a trial, that trial was excluded; however, if the participant felt comfortable completing them, subsequent trials were not excluded. The study was approved by the Institutional Review Board at Arizona State University (IRB ID: STUDY00000284).

## Procedures

Participants warmed up by engaging in five minutes of steady state aerobic exercise on an air resistance bike, an optional self-directed warm-up and/or stretch, and three warm-up sets of GM of 10 repetitions using only the 20 kg barbell (*Pescatello, 2013*). Thereafter, participants were given the option to warm up using additional weight before performing their 1RM estimation test. Using the methods described by *Baechle et al. (2008)*, each participant's 1RM was estimated by performing as many repetitions as possible with what each participant judged to be a moderately heavy load ($8.7 \pm 2.5$ repetitions with $55.5 \pm 25.4$ kg).

Participants were asked to wear appropriate clothing for access to the EMG sites. Before placing the electrodes on the skin, excess hair was removed with a razor, and skin was cleaned and abraded using an alcohol swab. Disposable, self-adhesive, Ag/AgCl pre-gelled, bipolar electrodes (Noraxon Product #272; Noraxon USA Inc., Scottsdale, AZ), with a diameter of 1 cm and an inter-electrode distance of 2 cm, were placed on the muscle bellies, parallel with muscle fibers of the right lateral hamstrings (LH), right medial hamstrings (MH), right lumbar erector spinae 3 cm lateral to spinous process L3 (LES),

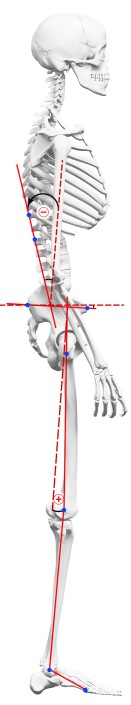

**Figure 1 Marker placement and angle calculations.**

and right thoracic erector spinae 5 cm lateral to spinous process T9 (TES) (*Konrad, 2005*; *McGill et al., 2009*).

Lumbar flexion was calculated by placing 25.4 mm spherical markers on T11, L1, and S2, and a 14.0 mm marker on the ASIS (*Kuo, Tully & Galea, 2009a*; *Kuo, Tully & Galea, 2009b*; *Kuo, Tully & Galea, 2009c*). In order to calculate hip, knee, and ankle angles, 14.0 mm markers were placed on the greater trochanter, lateral femoral epicondyle, lateral malleolus, and fifth metatarsophalangeal joint. The four-point angle created by the ASIS, PSIS, greater trochanter, and knee was subtracted from 90° to find the hip angle, similar to the methods used to calculate lumbar flexion (Fig. 1). Peak angles were recorded in the sagittal plane using a 120 Hz camera (Basler Scout scA640-120; Basler Vision Technologies, USA) and motion analysis software (MaxTRAQ 2D; Innovision Systems Inc., USA). Utilizing the maximal segmental flexion angles of the hip and knee, hamstring lengths were estimated relative to resting length, as per the equations and coefficients provided by *Hawkins & Hull (1989)*.

MVIC was taken for the erector spinae by performing a "superman"; in other words, having the participant lay prone and hyperextend their spine with their arms above their head. This position was chosen due to other data suggesting that a prone hyperextension elicits near-maximal erector spinae EMG (*Andersson et al., 1996*; *Callaghan, Gunning & McGill, 1998*; *Liefring et al., 1991*), and pilot testing suggested that, in trained individuals, the superman is just as effective as the methods described by *Vera-Garcia, Moreside & McGill (2010)*. Hamstrings MVIC was determined by having the participant lay prone and produce a force against manual resistance at 45° knee flexion and 0° hip flexion

(*Mohamed, Perry & Hislop, 2002*). In both MVIC positions, participants were instructed to contract "as hard as possible".

The barbell was placed on participants' upper trapezius, slightly above the level of the acromion (high bar position); arms were abducted and placed on the bar so the participants' elbows would not block the T11 marker upon descent; participants were instructed to stand how he normally would when performing the GM. Participants' shoes were removed before beginning. Participants performed one repetition of the GM using 50, 60, 70, 80, and 90% of estimated 1RM in a randomized order and were allowed two minutes rest between each repetition. No directions for depth or tempo were provided, as to let the participant perform the GM as he normally would in training.

EMG signals of the entire movement (eccentric and concentric), collected at 1,500 Hz, were analyzed using MyoResearch 3.4 (Noraxon, USA). A 10–500 Hz bandpass filter was applied to all EMG data. Peak and mean EMG data were rectified, smoothed using root mean square with a 50 ms window, and normalized to peak mean of a 100 ms window from the MVIC trial. While peak allows for all near-instantaneous increases in muscle activation to be seen, mean is robust to both movement artifact and time, thus providing a reliable average of EMG activity over the entire movement (*Renshaw et al., 2010*). EMG data presented as IEMG were rectified then integrated. IEMG was of particular interest because it has been linearly correlated with mechanical work (*Bouisset & Goubel, 1971*; *Bouisset & Goubel, 1973*), and is representative of the total electrical potential "spent" to complete the movement. Furthermore, because only one movement is being tested, IEMG is an appropriate measure (*Renshaw et al., 2010*). Kinematic data includes joint angles at the bottom of the eccentric of the movement, as judged by where maximal hip flexion occurred, which was determined by evaluating movement kinematics in MaxTRAQ (MaxTRAQ 2D; Innovision Systems Inc., USA).

## Statistical analyses

All data were normalized and entered into Stata 13 (StataCorp LP, College Town, TX), wherein one-way analyses of variance (ANOVA) with repeated measures were performed on all EMG and kinematic variables, with load being the independent variable. The normalization process was that described by *Loftus & Masson (1994)*, as to correct for between-participant variability, thus providing a truer representation of within-participant comparisons across trials. Tukey's HSD *post hoc* tests were employed on any measure that achieved a main effect. Alpha was set to 0.05 for significance. Partial eta-squared effect sizes were calculated and reported, as were their 95% confidence intervals (95% CI). Group means for each trial were used to calculate hamstrings length, utilizing the coefficients presented by *Hawkins & Hull (1989)*.

## RESULTS AND DISCUSSION

### Results

Participants ($n = 15$; age $= 24.6 \pm 5.3$ years; body mass $= 84.7 \pm 11.3$ kg; height $= 180.9 \pm 6.8$ cm) had $8.6 \pm 5.5$ years of weightlifting experience. During submaximal

**Peer**J

testing, participants completed $8.7 \pm 2.5$ repetitions with $55.5 \pm 25.4$ kg, and mean estimated 1RM was $76.0 \pm 24.9$ kg. All but one participant were able to complete every trial; the participant that failed to complete a repetition of 90% 1RM opted not to complete his 80% 1RM trial thereafter; however, his other trials were included. One participant's LH EMG was discarded due to abnormally high readings, indicative of an invalid signal, and one participant's 80% kinematic data was not included due to corruption of the motion capture video file.

### Mean EMG

Main effects were found for MH ($p < 0.001$; $F(4, 54) = 11.77$; partial $\eta^2 = 0.47$ (0.23, 0.58)), LH ($p < 0.001$; $F(4, 50) = 12.57$; partial $\eta^2 = 0.50$ (0.26, 0.61)), TES ($p < 0.001$; $F(4, 54) = 10.20$; partial $\eta^2 = 0.52$ (0.29, 0.62)), and LES ($p < 0.001$; $F(4, 53) = 6.08$; partial $\eta^2 = 0.31$ (0.08, 0.45)) mean EMG activity (Table 1).

### Peak EMG

Main effects were found for MH ($p = 0.016$; $F(4, 54) = 3.34$; partial $\eta^2 = 0.20$ (0.01, 0.33)) and TES ($p < 0.001$; $F(4, 54) = 7.12$; partial $\eta^2 = 0.35$ (0.11, 0.47)) peak EMG activity, but not LH ($p = 0.179$; $F(4, 50) = 1.64$; partial $\eta^2 = 0.12$ (0, 0.24)) or LES ($p = 0.659$; $F(4, 53) = 0.61$; partial $\eta^2 = 0.04$ (0, 0.12)) (Table 1).

### Integrated EMG

Main effects were found for MH ($p < 0.001$; $F(4, 54) = 6.31$; partial $\eta^2 = 0.32$ (0.09, 0.45)), LH ($p < 0.001$; $F(4, 50) = 11.99$; partial $\eta^2 = 0.49$ (0.25, 0.60)), TES ($p < 0.001$; $F(4, 54) = 8.36$; partial $\eta^2 = 0.38$ (0.14, 0.51)), and LES ($p < 0.037$; $F(4, 53) = 2.75$; partial $\eta^2 = 0.17$ (0, 0.30)) IEMG activity (Table 1 and Fig. 2).

**Table 1 Mean (95% CI) of measured EMG variables for each load performed.**

| | 50% 1RM | 60% 1RM | 70% 1RM | 80% 1RM | 90% 1RM |
|---|---|---|---|---|---|
| IEMG MH ($\mu V^* s$) | 255 (205, 304)[A] | 309 (261, 358)[AB] | 370 (290, 450)[AB] | 398 (262, 535)[AB] | 492 (301, 683)[B] |
| IEMG LH ($\mu V^* s$) | 201 (178, 224)[A] | 189 (151, 226)[A] | 242 (214, 270)[A] | 245 (222, 269)[A] | 337 (291, 382)[B] |
| IEMG TES ($\mu V^* s$) | 253 (214, 292)[A] | 284 (259, 309)[AB] | 317 (300, 333)[BC] | 358 (318, 384)[C] | 357 (326, 375)[C] |
| IEMG LES ($\mu V^* s$) | 373 (348, 397)[A] | 401 (365, 436)[AB] | 415 (375, 455)[AB] | 496 (401, 591)[B] | 474 (387, 561)[AB] |
| Mean MH (%MVIC) | 26.2 (23.1, 29.2)[A] | 28.4 (25.4, 31.4)[A] | 34.4 (31.4, 37.5)[B] | 37.3 (34.1, 40.4)[B] | 39.9 (36.8, 43.1)[B] |
| Mean LH (%MVIC) | 19.5 (17.2, 21.9)[A] | 19.3 (17.0, 21.7)[AB] | 24.2 (21.8, 26.6)[BC] | 26.4 (23.9, 28.8)[CD] | 30.4 (28.0, 32.9)[D] |
| Mean TES (%MVIC) | 46.4 (41.0, 51.9)[A] | 52.2 (46.7, 57.6)[AB] | 61.4 (56.0, 66.8)[BC] | 69.6 (63.9, 75.2)[C] | 66.6 (61.0, 72.2)[C] |
| Mean LES (%MVIC) | 50.8 (43.9, 58.0)[A] | 54.9 (47.9, 61.8)[A] | 61.5 (54.5, 68.5)[AB] | 73.1 (65.9, 80.4)[B] | 70.9 (63.5, 78.4)[B] |
| Peak MH (%MVIC) | 90.3 (80.5, 100)[A] | 94 (84, 104)[AB] | 113 (103, 123)[B] | 101 (90.6, 111)[AB] | 112 (101, 122)[B] |
| Peak LH (%MVIC) | 71.9 (64.6, 79.1) | 77.1 (69.9, 84.4) | 81.0 (73.8, 88.3) | 82.4 (74.9, 89.9) | 85.3 (77.8, 92.9) |
| Peak TES (%MVIC) | 122 (109, 135)[A] | 128 (115, 141)[A] | 145 (132, 158)[AB] | 158 (145, 172)[B] | 167 (154, 181)[B] |
| Peak LES (%MVIC) | 142 (125, 158) | 146 (130, 162) | 155 (138, 173) | 157 (141, 174) | 158 (141, 175) |

**Notes.**

[*] Loads sharing a letter are not statistically different ($p > 0.05$).

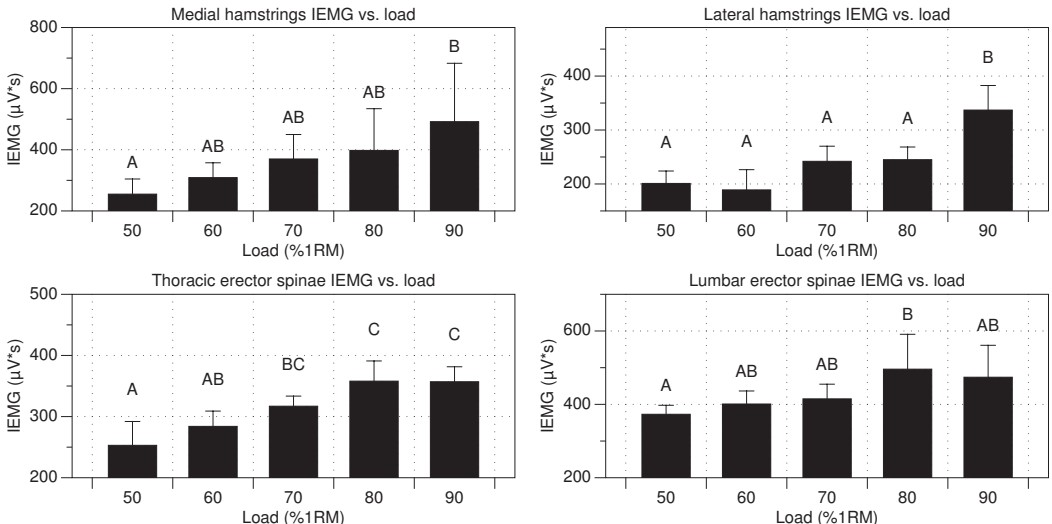

**Figure 2 IEMG activity versus load.** *Errors bars denote 95% CI. **Loads sharing a letter are not statistically different ($p > 0.05$).

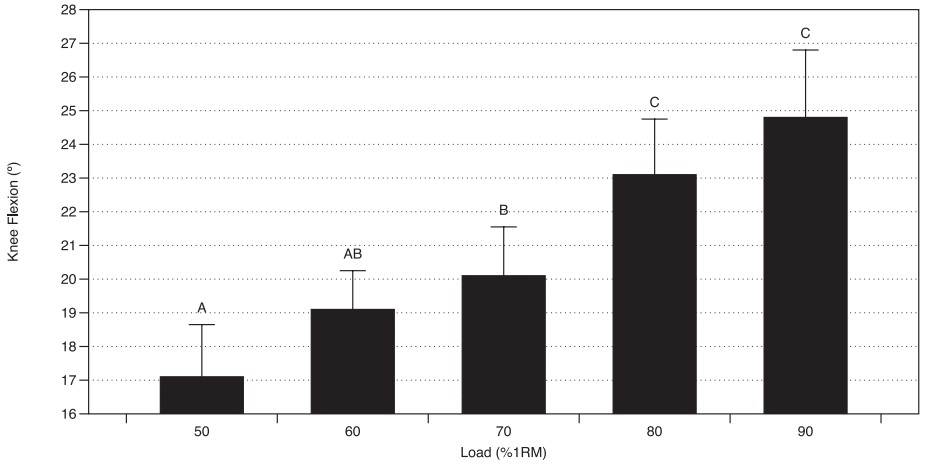

**Figure 3 Knee kinematics versus load.** *Errors bars denote 95% CI. **Loads sharing a letter are not statistically different ($p > 0.05$).

### Kinematics

Main effects were found for knee kinematics ($p < 0.001$; $F(4, 53) = 15.07$; partial $\eta^2 = 0.53$ (0.30, 0.63)), but not lumbar ($p = 0.172$; $F(4, 53) = 1.67$; partial $\eta^2 = 0.11$ (0, 0.23)), hip ($p = 0.715$; $F(4, 53) = 0.53$; partial $\eta^2 = 0.04$ (0, 0.11)), or ankle ($p = 0.184$; $F(4, 53) = 1.62$; partial $\eta^2 = 0.11$ (0, 0.22)) kinematics (Table 2 and Fig. 3).

Hamstring length was found to decrease with increases in load. All estimated lengths are greater than those found during sprinting (Table 3).

**Table 2** Mean (95% CI) of joint angle at movement depth for each load performed.

|  | 50% 1RM | 60% 1RM | 70% 1RM | 80% 1RM | 90% 1RM |
|---|---|---|---|---|---|
| Lumbar flexion (°) | 27.5 (25.7, 29.3) | 27.2 (25.6, 28.7) | 25.0 (23.0, 27.1) | 25.0 (23.6, 26.4) | 25.0 (22.5, 27.5) |
| Hip flexion (°) | 75.0 (73.6, 76.4) | 75.3 (74.3, 76.3) | 75.7 (74.8, 76.5) | 75.8 (74.5, 77.2) | 74.7 (73.2, 76.2) |
| Knee flexion (°) | 17.1 (15.6, 18.5)[A] | 19.1 (17.6, 20.5)[AB] | 20.1 (18.7, 21.5)[B] | 23.1 (21.5, 24.6)[C] | 24.8 (23.3, 26.3)[C] |
| Ankle plantar flexion (°) | 121.1 (120.0, 122.3) | 120.3 (119.6, 121.1) | 120.2 (119.4, 121.0) | 119.7 (119.0, 120.5) | 119.5 (118.1, 120.8) |

**Notes.**

Loads sharing a letter are not statistically different ($p > 0.05$).

**Table 3** Mean peak hamstring lengths normalized to resting hamstring lengths for each load performed.

|  | Sprint[a] | 50% 1RM | 60% 1RM | 70% 1RM | 80% 1RM | 90% 1RM |
|---|---|---|---|---|---|---|
| Biceps Femoris (LH) | 1.098 | 1.124 | 1.121 | 1.120 | 1.116 | 1.111 |
| Biceps Femoris (SH) |  | 0.997 | 0.996 | 0.995 | 0.993 | 0.992 |
| Semitendinosus | 1.082 | 1.126 | 1.123 | 1.123 | 1.117 | 1.112 |
| Semimembranosus | 1.075 | 1.108 | 1.105 | 1.103 | 1.097 | 1.091 |

**Notes.**

[a] *Thelen et al. (2004)*.

## Discussion

The findings of this study provide context as to how EMG and kinematic variables change as a function of load in the GM. Generally, IEMG of MH, LH, TES, and LES increased with load, except for the 80 and 90% trials, wherein TES and LES underwent an insignificant decrease (Table 1 and Fig. 2). The only kinematic variable that appeared to change with load was knee flexion, which increased with load, but not all increases were statistically significant (Table 2 and Fig. 3). The means of all other kinematic variables did not change more than three degrees and were, therefore, deemed unremarkable, especially considering the variability of the data (Table 2). Although the direction of the trends in mean and IEMG were similar, not all IEMG findings displayed similar relative changes in magnitude to the mean EMG data. This is not surprising, as IEMG includes a time component, whereas mean EMG data does not. In other words, a trial with a slightly lower mean may produce greater IEMG if trial duration is longer.

Like the findings of *Pinto et al. (2013)*, wherein investigators found that EMG activity plateaued from 80 to 90% of maximum voluntary isometric force production, participants' mean EMG activity of alleged prime movers (MH and LH) plateaued from 80 to 90% of 1RM (Table 1). However, significantly more IEMG activity is elicited in the LH at 90% than at 80%. These data suggest that, in the GM, heavier loads may provide a more potent training stimulus due to both an increase in total muscle activation and torque requirements.

The biceps femoris long head (BFLH) and semitendinosus (ST), respectively, are the two most commonly injured muscles of the hamstrings group (*Askling et al., 2007*; *De Smet & Best, 2000*), are subjected to the largest amount of stretch during the swing phase

of a sprint, and have the largest hip extension moment arms (*Chumanov, Heiderscheit & Thelen, 2006*; *Thelen et al., 2004*). Eccentric hamstring loading has been shown to be beneficial for decreasing the risk of hamstring injury in sport (*Askling, Karlsson & Thorstensson, 2003*; *Croisier et al., 2008*; *Heiderscheit et al., 2010*; *Petersen & Hölmich, 2005*). In this study, it was found that the BFLH and ST undergo the greatest stretch of all four hamstring muscles, respectively (Table 3), and decrease as a function of load. Additionally, these data show that the stretch in the GM is greater than the maximum stretch observed during a sprint, which were 1.098, 1.075, and 1.082 for the BFLH, SM, and ST, respectively (*Thelen et al., 2004*). Due to both the eccentric and stretching nature of the GM, it is postulated that the GM may be an effective exercise in preventing hamstring strains through the mechanisms described by *Brockett, Morgan & Proske (2001)*.

During the GM, the trunk is to be held stable as movement occurs about the hip joint, thus requiring an isometric action of the spinal erectors. However, these data, and others (*McGill et al., 2009*; *Schellenberg et al., 2013*), show that movement does occur in the lumbar spine (Table 2). Moreover, the lumbar erectors showed the greatest amount of mean EMG activity when normalized to MVIC (Table 1), although, this could be due to the MVIC position chosen and/or the ability of each participant to maximally contract his erector spinae muscles during MVIC testing. Nevertheless, lumbar extensors show an abnormally large potential for increases in strength with both isotonic and isometric exercise (*Graves et al., 1990*; *Graves et al., 1989*; *Pollock et al., 1989*); therefore, due to the high LES and TES EMG activity in the GM, the GM may be an effective training method for strengthening the erector spinae. However, training studies are needed to elucidate this effect.

Although statistical significance was found in these data, one cannot infer clinical significance, as what constitutes a clinically significant difference in EMG activity is not known. It is important to consider that the erector spinae MVIC position chosen in this investigation may not be a true MVIC position for everyone. Due to large inter-participant variability (*McGill, 1990*; *Vera-Garcia, Moreside & McGill, 2010*), it is challenging to find an MVIC position that leads to maximal activation for each subject; however, the superman exercise elicited greater EMG activity than the methods described by *Vera-Garcia, Moreside & McGill (2010)* in trained individuals during pilot testing. In another study, the superman exercise was only shown to elicit approximately 80% of MVIC EMG activity (*Ekstrom, Osborn & Hauer, 2008*), but this study did not use trained participants, nor were participants encouraged to maximally contract their erector spinae. Thirdly, having participants extend their arms on the barbell might have altered muscular activation. However, the experimenters noticed no change in form by having participants perform the GM in this manner, but this was not confirmed using kinematic data; all participants were comfortable with this position. As with any dynamic exercise, it is possible the motor units read by the surface EMG electrodes differed not only between loads, but also from MVIC trials. For example, the trials in which the hamstrings experienced greater lengthening, the electrodes may have detected a lesser number of motor units; moreover, it is possible that different motor units were detected during different trials. Additionally, the methods

described by *Baechle et al. (2008)*, which assume a linear association between loads lifted and repetitions performed, have not been validated in the GM. Furthermore, participants were not instructed to go to failure on every set, but doing so may have changed the results as this would have controlled for intensity of effort rather than just intensity of load (*Steele, 2013*), resulting in greater activation and more fatigue in all trials. Due to the size principle, EMG activity would theoretically have risen with each repetition until all motor units had been recruited and fatigued (*Carpinelli, 2008*), and light- versus heavy-load research supports this notion (*Sundstrup et al., 2012*). Lastly, the methods used to calculate hamstring length were not subject specific and only took into account sagittal plane movement.

This was the first study to examine both EMG and kinematic data as a function of load. In time, similar studies may show that certain loads may better mimic kinematic and EMG patterns in sport than other loads, or that certain loads emphasize some muscles more than others due to both changes in torque and EMG activity. It is recommended that future research examine these variables in addition to other kinetic variables, which will improve our understanding of the GM exercise and potentially allow coaches to improve program design strategies.

## CONCLUSIONS

The GM is a closed kinetic chain movement that is assumed to involve pure hip extension with a neutral spine. We have shown that the spine does not remain neutral during the GM exercise and in fact moves through flexion and extension. The hips move through a large degree of hip flexion, which requires a high degree of loaded stretch of the hamstring musculature. The hamstrings are of particular interest due to their propensity for injury (*Askling et al., 2007*; *De Smet & Best, 2000*), and due to the eccentric and stretching nature of the GM, it is postulated that the GM may be an effective movement for the prevention of hamstring injuries. It is clear that EMG activity of the alleged prime movers and knee kinematics are a function of load; therefore, it is recommended that coaches consider these variables within the realms of an athlete's entire program and load the athlete accordingly.

## ACKNOWLEDGEMENTS

We would like to thank Bryan Chung for his assistance and expertise which proved very valuable in the preparation of this paper.

### Funding
The authors declare there was no funding for this work.

### Competing Interests
The authors declare there are no competing interests.

## Author Contributions

- Andrew David Vigotsky conceived and designed the experiments, performed the experiments, analyzed the data, contributed reagents/materials/analysis tools, wrote the paper, prepared figures and/or tables, reviewed drafts of the paper.
- Erin Nicole Harper and Bret Contreras conceived and designed the experiments, reviewed drafts of the paper.
- David Russell Ryan performed the experiments, reviewed drafts of the paper.

## Human Ethics

The following information was supplied relating to ethical approvals (i.e., approving body and any reference numbers):

Arizona State University Institutional Review Board

IRB ID: STUDY00000284

## Supplemental Information

Supplemental information for this article can be found online at http://dx.doi.org/10.7717/peerj.708#supplemental-information.

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
