# Peer review of "Effects of load on good morning kinematics and EMG activity"

_PeerJ, doi:10.7717/peerj.708_

## Round 0.1 · original submission · Major Revisions

The reviewers find some interest in the study but both have substantial concerns that will require major revisions if we are to publish this paper. Please try to adjust the manuscript to use their constructive criticisms to improve the paper wherever possible, avoiding an emphasis on dialogue with the reviewers instead, but do provide a point-by-point Response document that addresses all their comments if you do submit a revised paper. The manuscript will be re-reviewed. Thank you for considering PeerJ.

·

Basic reporting

Upon reviewing the PeerJ policies it appears the authors have followed the primary requirement of submission.

Experimental design

In general I am comfortable with many aspects of the research design, but still the following need to be better explained or rationalized.
1. is the Baechle 1RM estimation approach likely to be valid for the good morning?
2. Why was the gluteus maximus not included as it is the primary uni-articular hip extensor muscle?
3. It is unclear why 90° was added to the four point angle to find the hip angle. Please make this clearer.
4. Please provide a reference for the MVIC approach of Superman exercise and why this was chosen over for example a back extension exercise?
5. Was a bandpass pass filter used to remove low and high frequency components of the EMG signal prior to data analysis and was the signal fullwave rectified prior to integration?
6. For the joint angles at the bottom of the movement do you mean the actual angles at the lowest position within the eccentric range?
7. Was the normality of the assessed? If not normal, meetings and interquartile ranges may be more appropriate than means and standard deviations. I would also suggest inferential statistics be it traditional based standard statistics or Hopkins based effect size statistics be used to determine differences between loads.

Validity of the findings

Results
The results presented so far are interesting but still need to be clarified in relation to my comments methodology. They would also be improved by performing inferential statistics stated previously as it is unclear the differences described in this section currently statistically or effect size different.
This section should clearly state where the figures are to be included in reference to the text and ensure these in numerical order.

Discussion
Previously you use the terms of medial and lateral head of the hamstrings, but here you use specific names of the hamstring muscles. Please be consistent in this approach.
The comparison between MVIC's of the lumbar erectors and the other muscles may not be valid as you even acknowledge that the MVIC approach for the lumbar erectors may not have approach true MVIC as closely as that of other muscles. I therefore suggest you more closely consider this section.

Additional comments

I like many aspects of this paper but feel additional detail in the provided within the methods and that inferential statistics need to be performed. Some limitations of EMG within studies of exercises may also need to be considered in relation to aspects methodology surrounding MVIC and also potential changes in range of motion and number of most units within the detection area of EMG electrodes across the various loads used in the study.

Reviewer 2 ·

Basic reporting

In general, the paper lacks depth. In particular, the authors make a number of comparisons with previous research but provide minimal detail from the previous papers and therefore there is a lack of transparency. The authors should provide more data from previous studies to assist with comparisons. Other examples of poor reporting include omission of text describing the hamstring length data in the results section and duplication of results in tables and figures.

Experimental design

There are multiple examples of details in the methods section that are too brief which limits reproducibility and comprehension by the reader. For example: reporting for EMG does not meet the full standards set by ISEK and details on the 2D analysis and calculation of hamstring length are also limited.

Validity of the findings

The argument provided in the discussion to omit the use of inferential statistics is not appropriate. Statistical tests to identify differences (i.e. Factorial ANOVA) are required. In addition, calculation of effect sizes is also recommended.

Additional comments

Introduction
Paragraph 1: The definition of specificity is vague (i.e. functional activities). Given the final sentence, the authors may wish to define specificity based on a mechanical perspective.
Line 20 to 22. Again, this has to be expanded to be made more clear. I.e. are you using kinematics and EMG as a measure of specificity, or is EMG used as a measure of intensity? This should be tied in more effectively with your definition of specificity in paragraph 1.
Line 26 to line 28. Data to provide context would be beneficial.
Line 29. Define relative muscular effort.
Line 32. Describe the good morning.
Line 33. Change building to increasing.
Line 32-34. Provide support for the statement.
Line 40. Given the previous comments regarding eccentric activity, it is perhaps beneficial to note that there is hip flexion and extension and state the muscle action corresponding to each motion.
Line 51. Provide data for lumbar erector spinae.
Line 48 to 55: More context has to be provided and it may be beneficial to provide the data from McAllister et al (2014) and compare with McGill et al (2009). The values provided are very low (due to unloaded nature), so it is not clear why this exercise would be effective in improving the strength of the back extensors.
Line 59. Remove "especially in the GM".
Line 60. Beneficial to remain objective unless you are stating a hypothesis. I.e. the purpose is not to show that load affects .... but to investigate if it does.
Line 63. Hamstring muscle length has to be introduced and its importance discussed in the introduction.

Methods
Paragraph 1. Justification is required for these hypotheses.
Paragraph 1. Beneficial to state how many sessions were used and the loading structure incorporated.
Line 73. Provide information on subjects (age, height, mass, training experience).
Line 92. Change bar to barbell
Line 96. Perhaps beneficial to provide the average number of repetitions and standard deviation produced in the final testing load.
Line 104-113. More detail is required for the 2D capture. Perhaps it would be beneficial to provide a diagram of the segments used and illustrate the angles.
Line 113. Briefly explain the principles behind the method outlined by Hawkins and Hull 1989.
Line 131. More detail is required regarding if the EMG was assessed over the entire repetition or just the concentric portion. Also, I think it would be beneficial to split the movement into eccentric and concentric portions and provide this information separately. Also, it is not clear why you are presenting IEMG data, when amplitude information is provided by the peak and mean EMG and importantly these data are normalized to minimize variation and provide more context for practitioners.
Line 133. Provide more detail, what variable was used to assess the bottom of the movement.
Statistics: Inferential statistics are required. Whilst for some of the EMG data patterns across loads are clear, it is expected that the null hypotheses are tested. In addition, it is important to provide effect sizes to provide a measure of differences.

Results
Remove duplication of results. The figures do not provide information additional to that provided in tables.
Line 137. One decimal place is more appropriate.
Line 144. As above, inferential statistics required to test this and also effect size calculation will provide a measure of the difference.
You need to report the results from the hamstring length data.

Discussion
Line 153. I disagree, these are two separate points. Inferential statistics can still be used to identify whether or not differences are likely to be the result of chance variation or underlying factors. In addition effect size calculation can provide a measure of how much of the variation in the data was due to the load used. Whether or not any true differences are meaningful is more difficult to ascertain. However, this difficulty does not nullify the importance of identifying true differences and providing a generalized measure of these differences.
Line 155. But the mean and peak EMG values in TES and LES do appear to increase. As mentioned previously, justification is required for the different EMG values used. If all three measures are kept then this discrepancy requires discussion.
Line 162. A more detailed comparison of the findings in the present study and that of Pinto et al (2013) would benefit the reader.
Line 176. Provide numerical values so the reader can compare.

---

## Round 0.2 · Major Revisions

Apologies for a slight delay but we obtained 1 review that is sufficient to convince me that the paper needs stronger data analysis (EMG filtering) and statistics- we recommend consulting with a statistician. Without those improvements we could not accept the paper. However, with substantial and serious revision we may be able to accept without further review.

·

Basic reporting

The manuscript reads well overall and I am happy with most of the changes made to this manuscript.

Experimental design

I still have some major concerns with the methodology.
The major issue being the lack of filtering of the EMG data. I suggest the authors consult the ISEK guidelines on EMG data collection and analysis. Without using a bandpass filter it is unknown accurate the EMG data actually is. While it may take some time, I see no reason why the authors can not go back to the raw EMG data, use a bandpass filter of something like 10-500 Hz before rectifying the data etc.
Further, I also believe strongly that the authors need to run some inferential statistics, be it effect sizes or ANOVAs on this data.

Validity of the findings

See above for the comment regarding the filtering of the data.

Additional comments

I like the overall approach of the data but without filtering of the EMG data or inferential statistics, I can not recommend acceptance of this paper.

---

## Round 0.3 · Minor Revisions

The final reviewer's comments are just requests for some tidying up and explaining, which I concur with. The Figures mainly do belong in Results, yes. Please do explain to the readers (briefly) the 3 EMG measures as recommended. Once this is done I will accept the paper. Apologies for the lengthy review process- we wanted to ensure that the paper is thoroughly inspected.

·

Basic reporting

Fine

Experimental design

Fine, but more clearly explain difference between mean, peak and IEMG in terms of their interpretations.

Validity of the findings

FIne, but have some questions on the difference in importance of mean, peak and IEMG. Please discuss in the methods what the difference in interpretation applies to these measures and continue this with more depth in the Discussion.

Additional comments

Much improved with EMG data analysis and statistical analysis worthy of publication now. Just some minor editing to finish.
Also, just double-check all revisions to ensure not typograohical erros etc.
I would also suggest Figures should be placed in Results not Discussion.

---

## Round 0.4 · accepted · Accept

The paper has clearly improved a lot from peer review and I can now very comfortably accept it-- congratulations!